# Associations between Sleep Disturbances, Personality Traits and Self-Regulation in a Sample of Healthy Adults

**DOI:** 10.3390/jcm13072143

**Published:** 2024-04-08

**Authors:** Ali Zakiei, Dena Sadeghi-Bahmani, Habibolah Khazaie, Zeinab Lorestani, Mohammad Sadeghi, Dariuosh Korani, Zeinab Sahraei, Saeid Komasi, Zeno Stanga, Annette B. Brühl, Serge Brand

**Affiliations:** 1Sleep Disorders Research Center, Kermanshah University of Medical Sciences, Kermanshah 6719851115, Iran; zakieiali@gmail.com (A.Z.); hakhazaie@gmail.com (H.K.); s_komasi63@yahoo.com (S.K.); 2Department of Psychology, Stanford University, Stanford, CA 94305, USA; bahmanid@stanford.edu; 3Department of Epidemiology and Population Health, Stanford University, Stanford, CA 94305, USA; 4Department of Clinical Psychology, Islamic Azad University of Kermanshah, Kermanshah 6719851115, Iran; z.lorestanei96@gmail.com (Z.L.); sadeghi.1996.mohammad@gmail.com (M.S.); dariuosh1356@gmail.com (D.K.); zeinab.sahraei65@gmail.com (Z.S.); 5Division of Diabetes, Endocrinology, Nutritional Medicine and Metabolism, University Hospital, University of Berne, 3010 Berne, Switzerland; zeno-giovanni.stanga@vtg.admin.ch; 6Centre of Competence for Military and Disaster Medicine, Swiss Armed Forces, 3008 Berne, Switzerland; 7Center for Affective, Stress and Sleep Disorders, Psychiatric Hospital of the University of Basel, 4002 Basel, Switzerland; annette.bruehl@upk.ch; 8Division of Sport Science and Psychosocial Health, Department of Sport, Exercise, and Health, Department of Medicine, University of Basel, 4052 Basel, Switzerland; 9School of Medicine, Tehran University of Medical Sciences, Tehran 1417466191, Iran; 10Substance Abuse Prevention Research Center, Kermanshah University of Medical Sciences, Kermanshah 6719851115, Iran; 11Center for Disaster Psychiatry and Disaster Psychology, Centre of Competence for Military and Disaster Medicine, Swiss Armed Forces, 4002 Basel, Switzerland

**Keywords:** self-regulation, sleep disturbances, personality traits

## Abstract

**Background:** Scientific evidence and everyday experience show that sleep disturbances and self-regulation as a proxy of stress reactivity are linked. Particular personality traits such as neuroticism, internalizing and externalizing problems are also associated with sleep disturbances. Here, we combined self-regulation and personality traits and associated these variables with subjective sleep disturbances. **Methods:** A total of 846 adults (mean age: 33.7 years; 78.7% females) completed questionnaires covering sleep disturbances, self-regulation and personality traits. **Results:** Higher scores for sleep disturbances were associated with higher scores for externalization, internalization, and instability and with lower scores for stability (all trait variables) and with poorer self-regulation (state variable). The regression model showed that higher scores for externalization and internalization (traits), and lower scores for self-regulation (state) predicted higher scores for sleep disturbance. Next, self-regulation had both a direct effect on sleep disturbance, and an indirect effect via personality traits. **Conclusions:** Sleep disturbances were related to both state (i.e., self-regulation) and trait (e.g., internalization and instability) dimensions. The current data analysis leapfrogs the state–trait dichotomy discussion and reconciles the state-and-trait approach in the prediction of poor sleep, though self-regulation appeared to have both direct and indirect effects on sleep disturbances.

## 1. Introduction

Insufficient sleep and sleep disorders are not only very common in the general population but also significantly associated with morbidity and mortality [1]. In adults, sleep durations that are either too long or too short are associated with a higher risk for cardiovascular disease, cognitive decline, coronary heart disease, falls, frailty, metabolic syndrome, and stroke, to name but a few [2]. Further, at a psychological level, poor sleep quality is associated with higher scores for impulsive behavior [3,4,5] including suicidal and non-suicidal behavior [6,7,8,9,10,11,12,13,14,15,16,17], and greater risk-taking [18,19,20], including risky driving behavior.

Another line of research has provided evidence that poor sleep is associated with poor emotion regulation and problematic emotional competencies [21,22,23,24,25,26,27,28,29,30,31,32,33,34,35,36,37], including higher rates for stress and dysfunctional coping [24,36,38]. Thus, while it appears plausible that poor sleep is associated with higher arousal [39,40,41] at both the psychological [41,42,43,44,45,46,47] and neurophysiological levels [27,37,39,40,48,49], testing the correlative and causal associations between sleep and emotion has proved methodologically more challenging [50,51]. Irrespective of methodological issues, however, virtually all psychiatric disorders and more specifically mood and anxiety disorders show a high co-occurrence with sleep disturbances [27,50,51,52,53,54]. As regards causal relationships, evidence is accumulating that symptoms of sleep disturbances precede mental health issues, including psychiatric disorders [52,53]. In complementary fashion, treatment of insomnia has been shown to prevent the emergence and onset of symptoms of depression [55]. There is also extensive evidence that individuals scoring high on stress, on stress reactivity [21], on daytime stress [36,56] and on dysfunctional coping also score high on insomnia [21,24,38,46,54,57,58]. Individuals scoring high on negative emotions and low on positive emotions were also those scoring high on insomnia [21]. In this respect and to make a case in point, previous research on emotional competencies among individuals with multiple sclerosis [59] showed that poor emotional competencies and higher scores for stress were associated with higher insomnia scores [60].

### 1.1. Insomnia and Personality Traits

While, conceptually, stress and coping are considered as states of reactivity in a specific context, dimensions of personality are understood as stable cognitive–emotional and behavioral traits. Dysfunctional personality traits increase the risk level for a broad variety of psychopathologies [61], and such traits do also have a negative impact on the treatment of psychiatric disorders, including insomnia [61,62,63]. For example, in one study [64] baseline introversion and neuroticism predicted higher scores for insomnia six months later. Complementary to this finding, individuals scoring high on insomnia have been found to be at greater risk of scoring high on personality alterations [65,66].

Two meta-analyses and systematic reviews have focused on the personality–sleep-link. In 2010, van de Laar et al. [67] summarized the results of 38 studies as follows: individuals scoring high on insomnia also score higher on symptoms of neuroticism, internalizing symptoms such as anxiety, depression, insecurity, and anxious concerns, including personality traits associated with perfectionism. At the time of this work no longitudinal data were available, so the direction of influence remained unclear: Did personality traits such as perfectionism, anxiety and neuroticism as a proxy for generalized cognitive–emotional framework of concerns and worry cause insomnia, or conversely did insomnia result in a generalized cognitive–emotional network of concerns and worry to counterbalance the sleep-related increase in daytime sleepiness, lack of vigilance and concentration, and diminished performance [67]? In 2023, Akram et al. [68] published a systematic review and meta-analysis of the results of 76 studies, including ten that were longitudinal. Higher scores for insomnia were associated with higher scores for neuroticism, introversion, perfectionism and elevated personality standards, including a negative affect, social inhibition and avoidance, impulsive behavior, including anger, hostility, and a negatively perceived self-concept. Results from the ten longitudinal studies suggested both directions of influence are possible, with insomnia leading to dysfunctional personality traits, and dysfunctional personality traits resulting in insomnia [68].

### 1.2. The Present Study

Acute bouts of stress, daytime stress and stress-reactivity [21,36,56], dysfunctional coping [21,24,38,46,54,57,58], and poor emotional competencies [60], including higher negative and lower positive emotions [21], have all been observed to trigger and maintain sleep disturbances. Further, cross-sectionally and longitudinally, sleep disturbances have been associated with problematic personality traits such as neuroticism, internalizing symptoms such as anxiety, depression, insecurity, and anxiety concerns.

Given this background, the aims of the present study were to investigate whether and if so to what extent both state (i.e., self-regulation) and personality traits (i.e., internalization) were associated with self-reported dimensions of sleep disturbances among a sample of adults from the general population.

We had two hypotheses and one research question. First, following previous research [21,24,36,38,46,54,56,57,58], we expected that lower scores for self-regulation would be associated with higher scores for sleep disturbances. Second, based on previous findings [67,68], we expected higher scores for unfavorable personality traits to be associated with higher scores for insomnia. Our research question was whether self-regulation impacted directly on sleep disturbances but also indirectly via personality traits.

## 2. Method

### 2.1. Study Design

In the period from April to June 2022, which is to say after the COVID-19 pandemic and its social restrictions, adults aged 18 to 65 years in Kermanshah province (western Iran) were asked to participate in this study. The study was advertised on the web pages of health organizations, public hospitals, universities and private companies. In addition, the study was promoted on social networking sites (SNS) such as Whatsapp^®^, Instagram^®^, Facebook^®^, Telegram^®^, and LinkedIn^®^.

Interested and eligible participants received a link to the online survey software of the Kermanshah University of Medical Sciences (http://digit.kums.ac.ir). The first page of the survey provided information about the purpose of the study and its secure and anonymous data management. Following this, participants signed a written informed consent form. To this end, they checked a box at the bottom of the first page to confirm that they had understood the study objectives, the anonymous management of data and the voluntary basis of participation in the study. By checking this box participants also confirmed they understood they could withdraw from the study at any time without any further justification. Once participants agreed to the terms of the study, they completed a series of questionnaires covering sociodemographic information, subjective sleep, self-regulation and personality traits (see below). On average, participants needed between 20 and 30 min to complete the questionnaire.

This study was registered at the Sleep Disorders Research Center of Kermanshah University of Medical Sciences in Iran and received approval from its ethics committee (IR.KUMS.REC.1399.1013, approved on 12 January 2021). The study was performed in accordance with the seventh revision [69] of the Declaration of Helsinki.

### 2.2. Participants

Inclusion criteria were as follows: 1. Being resident in Kermanshah province (western Iran) for at least five years; 2. age between 18 and 65 years; 3. willing and able to complete questionnaires written in Farsi/Persian; 4. compliance with the study conditions; and 5. checking the box for written informed consent. Exclusion criteria were as follows: 1. self-reporting chronic physical and mental health problems; 2. pregnant or breast feeding, given such states may alter current sleep and mood patterns.

A total of 900 participants responded to the online questionnaires, though 39 (4.34%) did not fully complete the survey and 15 participants (1.6%) were identified as “click-throughs” given they took less than seven minutes to complete the survey. Complete data were available for 846 participants.

### 2.3. Measures

#### 2.3.1. Sociodemographic Information

Participants reported on their age (years), gender at birth (male; female), civil status (single; married; or divorced), employment status (employed; unemployed), and highest educational level (below diploma; diploma; high school degree; or higher education).

#### 2.3.2. Sleep Quality

To assess sleep quality, participants completed the Farsi version [70,71,72,73] of the Pittsburgh Sleep Quality Index [74]. It is a validated inventory with 18 items grouped into 7 dimensions. The first dimension is related to subjective sleep quality assessed by a single question (Question 9). The next dimension is related to falling asleep, the score of which is based on two questions, namely the average score on item 2 and the score of part A of item 5. The third dimension concerns duration of sleep, assessed by one question. The fourth dimension is about the efficiency and effectiveness of a person’s sleep, the score of which is calculated by dividing the total number of hours of sleep by the total number of hours that a person is in bed, multiplied by 100. The fifth dimension is a measure of sleep disturbance obtained by calculating the average score on item 5. The sixth dimension, based on one question, is related to the use of hypnotic drugs. The last dimension concerns inappropriate performance during the day, assessed on the basis of two questions (average scores of items 7 and 8). Each question is scored from 0 to 3 and the score for each component is a maximum of three. The set of these seven dimensions form the total score of the scale, which ranges from 0 to 21. A higher score indicates more marked sleep disturbances (Cronbach’s alpha: 0.83). Further, Buyesse et al. [74] propose the following cut-off values: PSQI ≤ 5 points = good sleepers; PSQI > 5 points = poor sleepers. Based on these cut-off values, we reported the frequencies of poor and good sleeper.

#### 2.3.3. Self-Regulation

To assess self-regulation, participants completed the Farsi version [75] of the Self-Regulation Questionnaire (SRQ) [76]. The SRQ is a 31-item single-factor questionnaire that evaluates capacity for and degree of self-regulation. Typical items are as follows: “I can resist temptations”; “Little problems or distractions throw me off course.”; and “If I wanted to change, I am confident that I could do it”. Answers are given on five-point Likert scales ranging from 1 (=strongly disagree) to 5 (=strongly agree). Sum scores range between 31 and 155 points, with higher scores reflecting a higher degree of self-regulation (Cronbach’s alpha: 0.92).

#### 2.3.4. Personality Profiles

To assess personality profiles, participants completed the Farsi version [75] of the Affective and Emotional Composite Temperament Scale (AFECTS) [77]. This is a 60-item scale and evaluates the affective (12 items) and emotional (48 items) facets of temperament, separately. Dimensions are as follows: apathetic, depressive, hyperthymic, anxious, volatile, cyclothymic, obsessive, dysphoric, euthymic, disinhibited, irritable, and euphoric. Answers are given on five-point Likert scales ranging from 1 (=does not look like me at all) to 5 (=it looks exactly like me), with higher sum scores reflecting a more pronounced degree of the dimensions. Dimensions are clustered as follows: Internalization (sum of depressive + anxious + apathetic temperament); Externalization (sum score of euphoric + irritable + disinhibited temperaments]; Stability (sum score of obsessive + euthymic + hyperthymic temperaments); Instability (sum score of volatile + cyclothymic + dysphoric − euthymic temperaments) [77]; Cronbach’s alpha: 0.91. Next, a general score was calculated combining Internalization, Externalization, Instability and Stability. For Stability, scores were reversed so that a higher score reflected lower stability, and a lower score reflected greater stability.

#### 2.3.5. Data Analysis

Preliminary analysis: We tested whether there were systematic male–female differences with a series of *t*-tests. All ts were below 1.5 (all ps > 0.30). Gender was therefore not introduced as a confounder. Similarly, age was not significantly associated with any of the sleep, self-regulation, or personality trait dimensions (all rs < 0.10). Thus, age was not considered a possible confounder.

Associations between sleep, self-regulation and personality traits were examined with a series of Pearson’s correlations.

To predict sleep, a multiple regression analysis was performed. Preliminary conditions to run a multiple regression model were met [78,79,80]: N = 846 > 100; predictors explained the dependent variable (R = 0.481, R^2^ = 0.232), and the Durbin–Watson coefficient was 2.009, indicating that the residuals of the predictors were independent. Further, the variance inflation factors (VIF) were between 1.605 and 1.728; while there are no strict cut-off points for risk of multicollinearity, VIF < 1 and VIF > 10 indicate multicollinearity [78,79].

Next, the continuous variable of self-regulation was divided into the following categories: very low: 59–104.4; low: 104.5–114; medium: 114.1–121; high: 121.1–129.0; very high: 130–155. A multivariate ANOVA was performed with self-regulation categories as the independent factor (very low; low; medium; high; and very high) and sleep and personality traits (internalization; externalization; stability; and instability) as dependent variables. For F-tests, effect sizes were reported as partial eta-squared [η_p_^2^] and interpreted as follows: trivial (T) 0.019 < η_p_^2^, small (S) = 0.020 ≤ η_p_^2^ ≤ 0.059, medium (M) = 0.06 ≤ η_p_^2^ ≤ 0.139, or large (L) = η_p_^2^ ≥ 0.14 [81].

Last, to calculate the direct and indirect effects of self-regulation on sleep disturbances, we followed Rudolf and Müller [79], who proposed the following equation: the direct effect of self-regulation on sleep disturbances is r = x and β = y; the indirect effect of self-regulation on sleep disturbances via personality traits (total score) is r = x = β (y) + r_self-regulation on personality traits_ × β_personality traits on sleep disturbances_.

The level of significance was set at alpha < 0.05 (two-tailed). All statistical analyses were performed with SPSS^®^ 29.0 (IBM Corporation, Armonk, NY, USA) for Apple Mac^®^.

## 3. Results

### 3.1. General Sociodemographic Information

All statistical indices are reported in Tables and thus not repeated in the text.

Table 1 reports the general sociodemographic information.

The majority of participants were female and married, and had a diploma or above; half of the participants were either students or employed, and prevalently aged between 25 and 35 years.

### 3.2. Frequencies of Good and Poor Sleepers, Based on the Cut-Off Values of the Pittsburgh Sleep Quality Index (PSQI)

To identify good and poor sleepers, the following cut-off values have been proposed [74]: PSQI ≤ 5 points = good sleepers; PSQI > 5 points = poor sleepers. Based on these cut-off values, 492 (58.2%) participants were identified as good sleepers, and 354 (41.8%) were identified as poor sleepers.

### 3.3. Correlations between Sleep, Self-Regulation and Personality Traits

Table 2 reports the Pearson’s correlations between sleep, self-regulation and personality traits, including the descriptive statistical indices.

Higher scores for sleep disturbances were associated with lower scores for self-regulation, higher scores for internalization and externalization, higher scores for instability, higher scores for overall personality, and lower scores for stability.

Higher scores for self-regulation were associated with lower scores for internalization and externalization, instability, and a lower personality total score, and with higher scores for stability.

Higher scores for internalization were associated with higher scores for externalization, and instability, and a higher personality total score, and with lower stability.

Higher scores for externalization were associated with lower stability and higher instability, and with a higher personality total score.

Higher stability was unrelated to instability.

### 3.4. Predicting Sleep Disturbances

To predict sleep disturbances, a multiple regression analysis was performed with sleep disturbances as the dependent variable, and self-regulation and personality traits as predictors. Table 3 reports the model.

Lower scores for self-regulation and higher scores for externalization and internalization predicted higher sleep disturbances, while stability, instability and the personality total score were excluded from the equation as they did not reach statistical significance.

### 3.5. Categories of Self-Regulation on Sleep Disturbances and Personality Traits

In a further step, self-regulation scores were grouped into five categories: very low: 59–104.4; low: 104.5–114; medium: 114.1–121; high: 121.1–129.0; and very high: 130–155. Then, a multivariate ANOVA was performed with self-regulation categories as the independent factor (very low; low; medium; high; and very high) and sleep disturbances and personality traits (internalization; externalization; stability; instability; and personality total score) as dependent variables.

The results were as follows (see Table 4).

All F-tests were statistically highly significant (ps < 0.001; η_p_^2^ between 0.114 (medium effect size) to 0.423 (large effect size). The lower the category of self-regulation, the higher the sleep disturbances, the higher the scores for externalization, internalization, and instability, the lower the personality total score, and the lower the score for stability.

### 3.6. Direct and Indirect Effects of Self-Regulation on Sleep Disturbances

Figure 1 reports the statistics of the direct and indirect effects of self-regulation on sleep disturbances via personality traits.

The direct effect of self-regulation on sleep disturbances was r = −0.49; β = −0.17; the indirect effect of self-regulation on sleep disturbances via personality traits (total score) was r = −0.49 = β (−0.17) + r_self-regulation on personality traits_ = 0.686 × β_personality traits on sleep disturbances_ = 0.472.

## 4. Discussion

The aims of the present study were to investigate whether and to what extent sleep disturbances were associated with dimensions of both state (i.e., self-regulation) and trait (i.e., personality traits) among adults in the general population. Results showed that higher scores for sleep disturbances were associated with lower scores for self-regulation (state), lower scores for trait stability, and higher scores for trait externalization, trait internalization and trait instability. In addition, individuals scoring high on self-regulation also scored low on sleep disturbances both directly and indirectly via more favorable personality traits.

The present data add to the current literature in the following four ways. First, to predict sleep disturbances, we assessed dimensions of state and trait concomitantly; this allowed statistical evaluation of their relative contribution to the prediction of sleep disturbances. Results showed that low self-regulation, and higher externalization and internalization but not stability or instability predicted more marked sleep disturbances. Second and relatedly, we reconciled the state–trait dichotomy discussion in showing that both state and trait variables contributed to current sleep disturbances. However, third, it appeared that state dimensions predicted sleep disturbances, and the state of self-regulation further intensified such associations. Fourth and more specifically, low self-regulation magnified sleep disturbances both directly and indirectly via more problematic personality traits.

Two hypotheses and one research question were formulated. We consider each of these in turn.

### 4.1. State Self-Regulation and Sleep

Our first hypothesis was that lower scores for self-regulation would be associated with greater sleep disturbance and this was confirmed. Accordingly, the present results are in line with what has been observed in both clinical and non-clinical samples [21,24,36,38,46,54,56,57,58]. However, our findings expand upon the current literature in the following way: we showed that current state with respect to stress reactivity and poor coping with current stress was also associated with sleep disturbances in a general population sample.

### 4.2. Personality Traits and Sleep

Our second hypothesis was that higher scores for negative personality traits would be associated with higher scores for sleep disturbances and again this was confirmed. Given this, we further confirmed what has been reported in two major systematic reviews and meta-analyses [67,68]. However, the plus of the present study was that, unlike previous studies, we assessed both state and trait variables concomitantly, which allowed statistical evaluation of their relative contribution to sleep disturbances.

### 4.3. State–Trait Dimensions, and Direct and Indirect Effects on Sleep

Our research question concerned whether and if so to what extent degree of self-regulation impacted on sleep disturbances directly and/or indirectly via personality traits. First, as shown in Table 4, categories of self-regulation (very low to very high) systematically and linearly affected the intensity of sleep disturbances and personality traits, such that with increasing scores for self-regulation scores for sleep disturbance, instability, externalization and internalization decreased while stability increased. Second, and more importantly, the model for estimating the direct and indirect effects of self-regulation on sleep disturbances showed that higher scores for self-regulation resulted in lower scores for sleep disturbances both directly and indirectly via higher scores for dysfunctional personality traits. In our opinion, the merits of the model are two-fold: First, the model leapfrogs the state–trait dichotomy in showing that both state and trait variables contribute to sleep disturbances. Second, the model suggests that current mental state (here: self-regulation as a proxy for coping and stress reactivity) appears to impact directly on sleep disturbances, though a trait personality disposition (understood as a crystallized and callous coping strategy acquired over a person’s life time) further magnifies the effect of poor self-regulation on sleep disturbance.

The evidence available from this study is unable to shed any direct light on the psychophysiological mechanisms underlying the state–trait sleep links. In the absence of such evidence, we rely on previous models.

With respect to self-regulation, understood as successful coping or its absence, the hyperarousal model of insomnia [41] proposes that acute psychological stress, inadequate problem solving strategies, rumination and worrying lead to longer sleep onset latencies and more awakenings after sleep onset. At a physiological level, increases in monoamines, cortisol and orexin are adaptive changes to cope with a state of emergency and stress. In the event that psychological stress, inadequate problem solving, rumination and worrying persist, sleep alterations also persist and become chronic, with parallel physiological alterations.

For personality traits understood as crystallized, chronified and (dysfunctional) problem-solving strategies acquired over a person’s life, typically dimensions of neuroticism, anxiety, concerns, and worrying are associated with sleep disturbances, and it appears that the neurological correlate is increased cortical hyperarousal, at least among adolescents [39]. Similarly, Zhao et al. [49] observed alterations in cortical structural connectivity among individuals with insomnia when compared to individuals without insomnia. On this basis, we might claim that such neurological alterations would also be apparent among those participants scoring high on sleep disturbances and high on dysfunctional personality traits. Speaking more broadly about the associations between personality traits and coping with stress, among 70,652 adults sampled during the COVID-19 pandemic with its social restrictions, higher scores for stress were associated with being younger, female, and reporting more loneliness and distress, while personality traits such as extraversion and agreeableness were associated with better stress coping scores [82].

Despite the novelty of the results, the study has limitations. First, by default, cross-sectional study designs preclude testing of causal relationships and explanations; in the present study, we proposed that low self-regulation (along with dysfunctional personality traits) impacts on sleep disturbances, as proposed in the hyperarousal model of insomnia [41], though the opposite direction of influence is also entirely possible [21,52,53,54,83]. In this line, while we suggested that states impacted on sleep via traits, the reverse direction might be also likely, that traits could have affected states. A longitudinal design would also have allowed the exploration of potential bi-directional associations (see for instance for children and adolescents: [84,85]). Second, sleep was assessed subjectively. The alternative of, for example, sleep EEG recording would have allowed us to study self-regulation as a proxy of emotion regulation in relation to sleep stages in general and REM sleep more specifically, as reported elsewhere [27,34,36,56,86]. Third, and relatedly, the present study relied on self-reports; a thorough clinical interview could have shed light on additional psychological and psychiatric mental health issues such as major depressive disorder, anxiety, PTSD, and psychological trauma, including concussion [87] and mild traumatic brain injuries [88]. Consequently, it is conceivable that the present pattern of results was biased because of latent and unassessed psychological and neurophysiological factors, which might have distorted two or more variables in the same or opposite directions. Such latent variables could be a composite score of current mood, sleep patterns, social behavior and available cognitive–emotional resources [89]. Similarly, sampling via SNS should be considered as a further source of sampling bias.

## 5. Conclusions

In a sample of adults, self-regulation in terms of emotion regulation and stress reactivity, and personality traits of internalization, externalization and instability were associated with sleep disturbances. It also emerged that self regulation impacted on sleep disturbances both directly and indirectly via personality traits. At the practical level of treatment of sleep disturbances, two intervention strategies would appear to have potential. First, in the short term, a person might improve state dimensions such as coping with stress. Second, in the longer term, because personality traits are more resistant to change, a person might work on the personality traits and on core beliefs and values [90] and perfectionism [24].

## Figures and Tables

**Figure 1 jcm-13-02143-f001:**
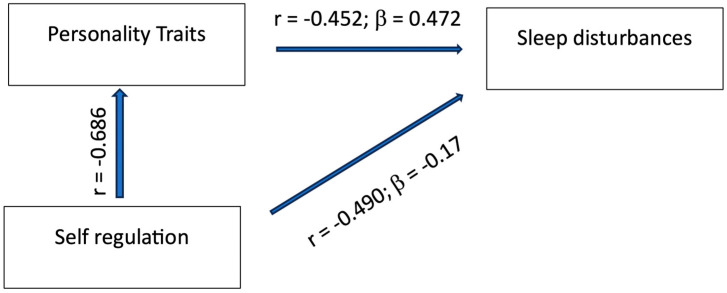
Direct effects of self-regulation on sleep disturbances, and indirect effects of self-regulation on sleep disturbances via personality traits.

**Table 1 jcm-13-02143-t001:** Sociodemographic information.

Variables		N (%)	Statistics
Sex	Male	180 (21.3)	X^2^ = 279.191, *p* < 0.001
	Female	666 (78.7)	
Education	Under diploma	157 (18.6)	X^2^ = 37.697; *p* < 0.001
	Diploma	246 (29.1)	
	Bachelor	264 (31.2)	
	Master and higher	179 (21.2)	
Marital status	Single	206 (24.3)	X^2^ = 598.730; *p* < 0.001
	Married	603 (71.3)	
	Other	37 (4.4)	
Occupation	Unemployment	432 (51.1)	X^2^ = 196.390; *p* < 0.001
	Student	103 (12.2)	
	Employment	311 (36.8)	
Age (Years)	18–25	102 (12.1)	X^2^ = 505.404, *p* < 0.001
	25–35	444 (52.5)	
	35–50	279 (33)	
	50–65	21 (2.5)	

**Table 2 jcm-13-02143-t002:** Descriptive statistics and Pearson’s correlation coefficients of and between sleep disturbances, self-regulation and personality traits.

	PSQI Total Score	Self-Regulation	Internalization	Externalization	Stability	Instability	Total Personality Score
PSQI total score	-	−0.49 ***	0.406 ***	0.401 ***	−0.219 ***	0.240 ***	0.452 ***
Self-regulation	-	-	−0.517 ***	−0.565 ***	0.516 ***	−0.346 ***	−0.686 ***
Internalization	-		-	0.566 ***	−0.206 ***	0.520 ***	0.845 ***
Externalization	-	-	-	-	−0.287 ***	0.275 ***	0.752 ***
Stability	-	-	-	-	-	−0.025	−0.500 ***
Instability	-	-	-	-	-	-	0.678 ***
Personality total score							-
M	5.63	117.08	8.01	6.44	10.04	7.08	29.46
SD	3.66	16.17	3.01	2.45	2.32	2.73	7.42

Notes: PSQI = Pittsburgh Sleep Quality Index. *** = *p* < 0.001.

**Table 3 jcm-13-02143-t003:** Multiple regression model to predict sleep disturbances as a function of self-regulation, externalization, internalization, stability and instability, and the personality total score.

Variables	Coefficients	Standard Error	β	t	*p*	R	R^2^	Durbin–Watson	VIF
Intercept	2.64	1.83		1.47	0.148	0.481	0.232	2.009	
Self-regulation	−0.039	0.010	−0.171	−3.89	<0.001				1.605
Externalization	0.265	0.059	0.178	4.54	<0.001				1.728
Internalization	0.233	0.051	0.192	4.53	<0.001				1.609
	Excluded variables: Stability; instability; personality total score: *p* < 0.35								

**Table 4 jcm-13-02143-t004:** Categories of self-regulation on sleep and personality traits.

Variables	Very Low	Low	Medium	High	Very High	F	*p*	η_p_^2^
	n = 169	n = 179	n = 164	n = 156	n = 178			
	M (SD)	M (SD)	M (SD)	M (SD)	M (SD)			
PSQI total	7.97 (4.25)	5.95 (3.25)	5.56 (3.25)	4.81 (3.21)	3.83 (2.82)	35.13	<0.001	0.144
Externalization	8.63 (2.51)	6.95 (2.16)	6.28 (1.82)	5.65 (1.88)	4.70 (1.88)	86.38	<0.001	0.292
Internalization	10.50 (2.79)	8.54 (2.39)	7.91 (2.66)	6.88 (2.57)	6.16 (2.69)	68.68	<0.001	0.247
Stability	8.32 (2.24)	9.54 (1.87)	10.11 (1.99)	10.64 (2.13)	11.59 (2.01)	60.24	<0.001	0.224
Instability	8.61 (2.92)	7.27 (2.42)	7.27 (2.61)	6.23 (2.24)	6.01 (2.60)	26.84	<0.001	0.114
Personality total score	37.40 (6.35)	31.23 (4.68)	29.35 (5.60)	26.12 (5.59)	23.29 (5.92)	153.23	<0.001	0.423

Notes: PSQI: Pittsburgh Sleep Quality Index; higher scores reflect a more disturbed sleep.

## Data Availability

Data are made available to explicit experts in the field upon request, upon well-formulated hypotheses and upon the explicit statement that data will not be misused and shared to third parties. Similarly, data are made available upon the explicit description of how data will be securely stored.

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
