# Peer review of "Associations between Sleep Disturbances, Personality Traits and Self-Regulation in a Sample of Healthy Adults"

_jcm, 2024, doi:10.3390/jcm13072143_

Round 1

Reviewer 1 Report

Comments and Suggestions for Authors

In this interesting study, based on a sample of 846 adults, the authors investigated the associations between sleep disturbances, personality traits and self-regulation. The Authors concluded that subjective sleep disturbances were related to both state (i.e., self-regulation) and trait (for instance, internalization and instability) dimensions. The study is well-designed, with a comprehensive self-rating questionnaires panel and a thorough statistical analyses plan. Results are reported accurately and discussed in a polished manner.

I have some minor suggestions that may further increase the quality of this study article:

Title/Abstract should specify that the study was conducted in a healthy population.

In the Data analysis subsection (LL 218-220), the Authors state that “the continuous variable of self-regulation was categorized into the following categories: very low: 59-104.4; low: 104.5-114; medium: 114.1-121; high: 121.1-129.0; very high: 130-155”. Is this a standard categorization? If so, the Authors may provide a reference. 

Considering that the COVID pandemic has affected the way stress and personality traits influence coping mechanisms (Prerna Tigga and Garg, 2022), I suggest that the Authors add a brief comment on this issue.

Among the limitations of this study, the sampling bias arising from the diffusion of the survey through social network and chats should be mentioned. 

References

Prerna Tigga N, Garg S. Prediction of Global Psychological Stress and Coping Induced by the COVID-19 Outbreak: A Machine Learning Study. Alpha Psychiatry. 2022;23(4):193-202. doi: 10.5152/alphapsychiatry.2022.21797. 

Comments on the Quality of English Language

/

Author Response

We thank Reviewer # 1 for the valuable and encouraging comments, which helped us to improve the quality of the revision. Please find the detailed point-by-point-response attached. 

Thank you again for all your kind efforts. 

Reviewer 2 Report

Comments and Suggestions for Authors

Questions:

1.     Line 205: (ala ps > .30)  Please edit. Do you mean (all P-levels were > .30)?

2.     Line 207: (all rs <.10)  Please edit. Do you mean (all correlation coefficients were < .10)?

3. The chi-square values in Table 1 are unclear, and they are not needed.

4. How did you verify the nature of the distributions? Were they normal? If they were not normal, the Pearson correlations are not ideal.

5.     As there are many different correlations, it is advised to use the Bonferroni transformations. Without that, the P-values have little meaning.

6.     Tables 3 and 4 are difficult to understand. Suppose that the reader is a clinician who is not familiar with regression analyses and different coefficients. Try to make them easier to understand. One possibility is that you look for a pathological cutpoint of the PSQI. One possible value could be > 5 for suspected poor sleepers. Then, you could use logistic regression analysis, which is easier to understand.

7. In the models, if gender and age may not have significant effects, you should show that in a model. Also, the model should include other possible confounders like education, profession, etc (see your demographics).

8. Ideally, you may give univariate odds ratios, for example, for PSQI global score > 5, and other models.

9.     In the discussion, please use subtitles.

10.  In the limitations sections, you should also discuss the limitations related to the sampling. Does your sample represent the general population? Probably not as, for example, elderly people do not commonly use internet media. Please discuss.

11.  What is the external validity of your results?

12.  Finally, please write more about your study's clinical and practical implications. How can a physician or a nurse use your results? How to interpret the results?

Comments on the Quality of English Language

Please edit the language carefully.

Author Response

We thank Reviewer # 2 for the valuable and encouraging comments, which helped us to improve the quality of the revision. Please find the detailed point-by-point-response attached. 

Thank you again for all your kind efforts. 
